# Anti-Inflammatory and Antifungal Activities of Wood Essential Oil from *Juniperus morrisonicola* Hayata

**DOI:** 10.3390/plants14182924

**Published:** 2025-09-20

**Authors:** Nai-Wen Tsao, Shih-Chang Chien, Yen-Hsueh Tseng, Sheng-Yang Wang

**Affiliations:** 1Program in Specialty Crops and Metabolomics, Academy of Circular Economy, National Chung-Hsing University, Nantou 540, Taiwan; qmotkd@smail.nchu.edu.tw; 2Experimental Forest Management Office, National Chung-Hsing University, Taichung 402, Taiwan; 3Taiwan Forestry Research Institute, Taipei 100, Taiwan; 4Department of Forestry, National Chung-Hsing University, Taichung 402, Taiwan; 5Agricultural Biotechnology Research Center, Academia Sinica, Taipei 108, Taiwan

**Keywords:** *Juniperus morrisonicola* Hayata, essential oil, sesquiterpenoids, cedrol, widdrol, thujopsene, anti-inflammatory, cytotoxicity, antifungal activity

## Abstract

This study presents the first comprehensive analysis of the wood essential oil from *Juniperus morrisonicola* Hayata (Jm-EO), an endemic conifer in Taiwan. Gas chromatography–mass spectrometry (GC-MS) revealed a sesquiterpenoid-rich profile, with cedrol, widdrol, and thujopsen comprising over 55% of the total essential oil content. Jm-EO exhibited significant anti-inflammatory activity in vitro, notably inhibiting nitric oxide production in lipopolysaccharide (LPS)-stimulated RAW264.7 macrophages (IC_50_ = 12.9 μg/mL). Among the major constituents, widdrol demonstrated the most potent anti-inflammatory activity (IC_50_ = 24.7 μM), followed by thujopsene and cedrol, representing the first report of widdrol’s anti-inflammatory activity. Jm-EO also showed cytotoxic effects against HepG2 hepatocellular carcinoma cells (IC_50_ = 41.5 μg/mL at 48 h) and achieved complete inhibition of *Laetiporus sulphureus* at 100 ppm. These findings suggest that Jm-EO is a promising natural resource with potential applications in anti-inflammatory drug development and as an eco-friendly wood preservative.

## 1. Introduction

*Juniperus morrisonicola* Hayata, an endemic coniferous species of Taiwan, belongs to the Cupressaceae family and grows at elevations of 3200 to 3990 m along the Central Mountain range. It is one of the highest-altitude conifers; it displays remarkable morphological diversity, ranging from tall trees to dense, shrub-like forms on mountain summits. Its wood is densely grained and aromatic, traditionally used for incense, and valued for its potential for ornamental landscaping and bonsai cultivation.

Wood essential oils from *Juniperus* species exhibit diverse, species-specific chemical profiles, typically characterized by unique combinations of sesquiterpenoids and monoterpenoids. For example, *Juniperus oxycedrus* wood essential oil is rich in sesquiterpenes such as *δ*-cadinene, *cis*-thujopsene, and *α*-muurolene, and demonstrated strong antioxidant, hypoglycaemic, and antiproliferative activities [1,2]. Similarly, the heartwood essential oils of *Juniperus virginiana*, *Juniperus occidentalis*, and *Juniperus ashei* contain distinctive constituents with documented wound-healing and anti-inflammatory properties [3]. Essential oils from various *Juniperus* species also display antimicrobial activity against pathogens such as *E. coli* and *S. aureus*, underscoring their broad therapeutic potential [4,5]. In addition, Texas cedarwood oil (*Juniperus deppeana*) is commercially valued for its fragrance, although its full chemical profile remains inadequately characterized [6].

These findings highlight the pharmaceutical, cosmetic, and industrial relevance of *Juniperus* essential oils.

Despite extensive research on essential oils from various *Juniperus* species, the phytochemical composition and bioactivities of *J. morrisonicola* remain largely uncharacterized. To address this gap, the present study provides a comprehensive analysis of the *J. morrisonicola* wood essential oil using gas chromatography–mass spectrometry (GC-MS), alongside evaluations of its biological activities, including anti-inflammatory effects in macrophages, cytotoxicity against hepatocellular carcinoma cells, and antifungal activities against wood-decay fungi. These results not only expand our understanding of the unique phytochemical traits of this endemic Taiwanese conifer but also underscore its potential value in pharmaceutical and industrial applications.

## 2. Results and Discussion

### 2.1. Chemical Composition of the Essential Oil

The chemical composition of the essential oil extracted from *J. morrisonicola* (Jm-EO) wood was analyzed by GC-MS (Figure 1), and the results are summarized in Table 1. A total of 15 compounds were identified, accounting for 92.56% of the total oil composition. The main constituents identified were cedrol (22.87%), widdrol (20.82%), and thujopsene (12.26%), together representing 55.95% of the total essential oil. Other notable components included 8,14-cedranoxide (9.50%), *α*-cedrene (8.16%), *α*-selinene (4.31%), *β*-himachalene (4.12%), and zingiberene (3.09%). Notably, many of the identified compounds possess a cedrane-type sesquiterpene skeleton—such as cedrol, *α*-cedrene, *β*-cedrene, and 8,14-cedranoxide—suggesting that Jm-EO exhibits characteristic cedar-like aromatic and chemical properties.

Compared to wood essential oils from other *Juniperus* species, Jm-EO exhibits a distinct chemical profile. Species such as *Juniperus communis* and *Juniperus chinensis* typically contain monoterpenes like *α*-pinene, sabinene, and limonene [7,8], which are notably absent in Jm-EO. This absence underscores its unique sesquiterpenoid-dominant composition. A similar sesquiterpenoid-rich profile has been observed in *Juniperus thurifera*, particularly with levels of cedrol, thujopsene, and widdrol, albeit in varying proportions [9]. Likewise, essential oil from *Juniperus foetidissima* contains elevated levels of cedrol and widdrol [10], supporting the chemotaxonomic similarities within the *Juniperus* genus. These comparisons demonstrate that Jm-EO maintains a unique yet phylogenetically related chemical signature characterized by dominant sesquiterpenoids.

The major constituents—cedrol, a tricyclic sesquiterpenoid, and widdrol, a bicyclic sesquiterpenoid with a distinct 6-7 ring system—are known for their diverse biological activities. Cedrol has been reported to exhibit a broad spectrum of pharmacological effects, including anti-inflammatory [11,12], anticancer [13,14,15,16], anti-obesity [17], antifungal [18], and anti-melanogenic [19] effects. Additionally, cedrol has shown potential in alleviating cerebral ischemia [20,21] and has demonstrated analgesic [22], neuroprotective [23], and hair growth-promoting effects [24,25,26]. More recently, we found cedrol enhances ATP content and the trans-epithelial electrical resistance (TEER) value in the intestinal epithelial barrier. Moreover, cedrol mitigates the IC-induced decrease in mRNA expression of Tight junction proteins (ZO-1, Occludin, and Claudin-1), thereby ameliorating intestinal epithelial barrier dysfunction [27]. Similarly, widdrol has been associated with anticancer [28], anti-obesity [29], anti-angiogenic [30], antifungal [31], and anti-melanogenic properties [29]. Therefore, the high abundance of these bioactive components may underlie the diverse biological effects of Jm-EO and lay a solid foundation for further functional and pharmacological investigations.

### 2.2. Anti-Inflammatory Activity of Jm-EO

The anti-inflammatory activity of Jm-EO was assessed by measuring its inhibition of NO production in LPS-stimulated RAW 264.7 macrophage cells (Figure 2). Prior to this, cytotoxicity was evaluated to determine non-toxic concentrations and ensure that observed effects were not attributed to cell death.

Jm-EO maintained high cell viability (>90%) at concentrations up to 20 μg/mL (Figure 2a), while viability sharply declined to approximately 2% at 40 μg/mL. For comparison, curcumin—the positive control—retained ~95% viability at 2.5 μg/mL but exhibited moderate cytotoxicity (~65% viability) at 5 μg/mL.

Jm-EO demonstrated a dose-dependent NO inhibition in activated macrophages (Figure 2b). Minimal inhibition was observed at 1 μg/mL, but NO production was suppressed by ~15%, 40%, and 80% at 5, 10, and 20 μg/mL, respectively. The calculated IC_50_ for NO inhibition was calculated at 12.9 μg/mL, indicating notable anti-inflammatory potential. At 20 μg/mL, Jm-EO showed comparable inhibitory activity to curcumin at 2.5 μg/mL.

This observed activity is likely attributable to major constituents of cedrol (22.87%), widdrol (20.82%), and thujopsene (12.26%), which together account for over 55% of the oil (Table 1). Prior research has reported anti-inflammatory effects of cedrol, including suppression of pro-inflammatory cytokines and modulation of signaling pathways [32]. The potent NO inhibition by Jm-EO at non-cytotoxic doses highlights its potential as a natural anti-inflammatory agent. Although curcumin demonstrated higher potency at lower concentrations, the comparable efficacy of Jm-EO at 20 μg/mL suggests that further purification or isolation of active constituents from *J. morrisonicola* could yield compounds with enhanced anti-inflammatory activity.

### 2.3. Anti-Inflammatory Activity of Major Components

After evaluating the whole essential oil, the anti-inflammatory potential of its three major components—cedrol, widdrol, and thujopsene—was further investigated. Their cytotoxicity and inhibitory effects on NO production were assessed in RAW 264.7 cells across a concentration range of 5–100 μM (Figure 3).

The cytotoxicity profiles of the three compounds varied distinctly (Figure 3a). Cedrol maintained high cell viability (>85%) up to 50 μM, decreasing to ~65% at 100 μM. Widdrol displayed a clear concentration-dependent cytotoxicity, with viability gradually decreasing from nearly 100% at 5 μM to ~60% at 100 μM. Thujopsene exhibited moderate cytotoxicity, with viability remaining above 80% up to 50 μM, but dropping sharply to about ~20% at 100 μM. Accordingly, concentrations ≤ 50 μM were selected for subsequent anti-inflammatory assays to avoid cytotoxic interference.

As shown in Figure 3b, all three compounds inhibited NO production in a concentration-dependent manner. However, the strong suppression observed at 100 μM likely reflects cytotoxicity effects—particularly for thujopsene—rather than genuine anti-inflammatory activity. Therefore, assessments at non-cytotoxic concentrations (≤50 μM) offer a more accurate evaluation of their anti-inflammatory potential. To compare potencies, the half-maximal inhibitory concentration (IC_50_) for NO production was calculated for each compound. Widdrol exhibited the greatest potency (IC_50_ = 24.7 μM), followed by thujopsene (IC_50_ = 30.3 μM), while cedrol was the least potent (IC_50_ = 41.1 μM). These values indicate that widdrol is the most effective NO inhibitor among the three major components tested.

The differences in anti-inflammatory potency among these compounds may be attributed to their structural characteristics. Notably, widdrol—characterized by a bicyclic sesquiterpenoid framework with a distinct 6-7 fused ring system—exhibited the strongest activity (lowest IC_50_ value). To the best of our knowledge, this is the first report of widdrol’s anti-inflammatory activity. This novel finding suggests that specific structural features of widdrol might underlie its biological activity and highlights its potential as a lead compound for therapeutic development.

These results indicate that the anti-inflammatory activity of Jm-EO likely arises from the combined or synergistic effects of its major components, with widdrol appearing to be a particularly significant contributor. Based on both its high abundance in the essential oil (20.82%, Table 1) and its superior inhibitory potency (lowest IC_50_ = 24.7 μM), widdrol is speculated to be the primary contributor to the overall anti-inflammatory activity of the essential oil.

These findings suggest that the anti-inflammatory activity of Jm-EO likely results from the combined or synergistic effects of its major constituents, with widdrol emerging as a particularly significant contributor. Given its high abundance in the essential oil (20.82%, Table 1) and its superior inhibitory potency (lowest IC_50_ = 24.7 μM), widdrol is presumed to be the primary driver of the observed anti-inflammatory effects.

### 2.4. Cytotoxic Effects of Jm-EO on HepG2 Cells

The cytotoxic effects of Jm-EO on HepG2 human hepatocellular carcinoma cells are shown in Figure 4. Cell viability was assessed after treatment with Jm-EO at concentrations ranging from 5 to 80 μg/mL for 24 and 48 h, using plumbagin as a positive control. Jm-EO exhibited dose- and time-dependent cytotoxicity, with stronger effects observed at higher concentrations and longer exposure durations. Notably, treatment with 40 μg/mL of Jm-EO reduced cell viability to 66% at 24 h and further to 36% at 48 h. The highest concentration tested (80 μg/mL) resulted in a dramatic reduction in viability to approximately 5% at both time points, indicating marked cytotoxicity.

As anticipated, plumbagin showed strong cytotoxicity: 1 μM decreased cell viability to 84% (24 h) and 66% (48 h), while 2 μM reduced viability to 10% and 5%, respectively. Based on these results, the IC_50_ of Jm-EO against HepG2 cells after 48 h was estimated at approximately 41.5 μg/mL.

The observed cytotoxicity of Jm-EO may be largely attributed to its major sesquiterpenoid constituents, including cedrol, widdrol, and thujopsene. These compounds have been previously reported to exert anticancer effects through mechanisms such as induction of apoptosis, cell cycle arrest, and modulation of critical cellular signaling pathways [13,14,15,16,28]. Further studies elucidating the precise molecular pathways involved in the cytotoxic effects of these compounds on HepG2 cells are warranted to support their potential for therapeutic development.

### 2.5. Antifungal Activity of Jm-EO

Table 2 presents the antifungal effects of Jm-EO against four wood-decay fungi: *Fomitopsis pinicola*, *Trametes versicolor*, *Lenzites betulina*, and *Laetiporus sulphureus*. Jm-EO exhibited concentration-dependent inhibition across all tested species, though the degree of susceptibility varied. A 25 ppm moderate inhibition was observed against *F. pinicola* (27.68%), while *L. sulphureus* (17.73%), *T. versicolor* (13.42%), and *L. betulina* (10.70%) were less affected, with inhibition rates ranging from 12.36% to 38.41%.

Remarkably, Jm-EO completely inhibited the growth of *L. sulphureus* at 100 and 200 ppm, indicating the high sensitivity of this brown-rot fungus. In contrast, the inhibition of other fungi increased more gradually, reaching 63.93% for *F. pinicola*, 44.34% for *L. betulina*, and 35.88% for *T. versicolor* at 200 ppm.

The observed differences in fungal susceptibility suggest potential species-specific modes of action. The strong inhibitory effect against *L. sulphureus*, a major brown-rot agent in wood structures, is particularly noteworthy. This activity is likely attributable to the high content of sesquiterpenoids in Jm-EO, especially cedrol, widdrol, and thujopsene. Previous studies have shown that cedrol can induce apoptosis in brown root rot fungi via reactive oxygen species generation [18].

These findings highlight Jm-EO as a promising natural antifungal agent, particularly for wood preservation against *L. sulphureus*. Further investigation into its mode of action and formulation development may support its application as an eco-friendly alternative to synthetic fungicides.

This study focused on evaluating the whole essential oil to establish its overall antifungal potential, given that synergistic interactions among components often contribute to bioactivity in natural products. While this approach provides practical insights for direct application, examination of individual compounds based on literature evidence offers important mechanistic insights. Among the major constituents identified, cedrol (22.87%) has well-documented antifungal properties, with previous studies demonstrating its ability to induce apoptosis in fungal cells through ROS generation [15] and its effectiveness against wood-decay fungi at 100 μg/mL [33,34]. In contrast, thujopsene (12.26%) exhibits only weak antifungal activity [35], while widdrol (20.82%), despite its high abundance, has no reported antifungal activity in the literature. Based on these findings, we hypothesize that the observed antifungal activity of Jm-EO is primarily attributed to its high cedrol content. However, considering the concentration calculations, the actual cedrol concentration in our tested essential oil may be insufficient to fully account for the observed antifungal effects when compared to literature values. This suggests that widdrol (20.82%), despite lacking reported antifungal activity in the literature, may contribute to the overall antifungal efficacy, either through direct antifungal activity or synergistic interactions with cedrol. Nevertheless, systematic evaluation of individual compounds and their potential synergistic interactions remains essential for developing optimized antifungal formulations and understanding the complete mechanism of action.

## 3. Materials and Methods

### 3.1. Plant Material

In 2017, wood of *J. morrisonicola* Hayata, which was taken from an approximately 500-year-old tree with a diameter at breast height of about 20 cm, was collected on the summit of Mt. Jade (Yushan), Taiwan. Heartwood and sapwood were not distinguished during sampling. Professor Yen-Hsueh Tseng (National Chung Hsing University) authenticated the plant material, and a voucher specimen has been deposited in the herbarium of the Department of Forestry, National Chung Hsing University.

### 3.2. Essential Oil Extraction and GC-MS Analysis

The essential oil from *J. morrisonicola* was extracted by steam distillation. Cut into small pieces, 250 g of fresh wood was placed in a Clevenger-type apparatus with 1.6 L of double-distilled water. The wood was steam-distilled for 6 h. The extracted essential oil was collected in about 5.3 mL, a yield of 2.1% (mL/w), and stored in airtight amber vials at 4 °C until further analysis. The chemical composition of *J. morrisonicola* wood essential oil (Jm-EO) was analyzed using gas chromatography–mass spectrometry (GC-MS). GC-MS analysis was performed using an ITQ 900 mass spectrometer equipped with a DB-5MS capillary column (30 m × 0.25 mm i.d., 0.25 μm film thickness). The GC oven temperature program was set as follows: initial temperature 40 °C, then increased at a rate of 4 °C/min to 180 °C, followed by an increase at 15 °C/min to 280 °C, and held for 5 min. Other operating parameters were as follows: injector temperature, 240 °C; ion source temperature, 250 °C; electron impact ionization at 70 eV; helium as carrier gas at a flow rate of 1 mL/min; injection volume, 1 μL (1% solution in ethyl acetate) with a split ratio of 1:200; and mass scan range, 50–600 m/z. The compounds were identified by comparing their mass spectra with National Institute of Standards and Technology (NIST) 2020 GC-MS libraries. Kovats indices (KIs) were calculated for all compounds using a series of *n*-alkanes (C9-C30) under the same chromatographic conditions. Authentic standards further confirmed the major components, cedrol (Sigma-Aldrich, St. Louis, MO, USA), widdrol (RR Scientific, Irwindale, CA, USA), and thujopsene (Cayman Chemical, Ann Arbor, MI, USA). The relative percentage of each component was calculated based on the GC-MS peak area.

### 3.3. Cell Culture

RAW 264.7 murine macrophage cells and HepG2 human hepatocellular carcinoma cells were obtained from the Bioresource Collection and Research Center (BCRC, Hsinchu, Taiwan). RAW 264.7 cells were cultured in Dulbecco’s Modified Eagle’s Medium (DMEM) supplemented with 10% fetal bovine serum (FBS), 1 mM sodium pyruvate, 1% glutamax, and 1% penicillin–streptomycin. HepG2 cells were maintained in DMEM supplemented with 10% FBS and 1% penicillin–streptomycin. Cells were incubated at 37 °C in a humidified 5% CO_2_ atmosphere. The culture medium was changed every 2–3 days, and cells were subcultured at 80–90% confluence using 0.25% trypsin–EDTA solution.

### 3.4. Anti-Inflammatory Activity Assay

The anti-inflammatory activity was assessed by measuring inhibition of nitric oxide (NO) production in lipopolysaccharide (LPS)-stimulated RAW 264.7 cells following the established protocols for essential oil bioactivity evaluation [36]. RAW 264.7 cells were cultured in DMEM supplemented with 10% fetal bovine serum and maintained at 37 °C in a 5% CO_2_ humidified incubator. For the assay, cells were seeded in 96-well plates at a density of 1 × 10^5^ cells/well and allowed to adhere overnight. The cells were then treated with various concentrations of Jm-EO (1, 5, 10, 20, and 40 μg/mL) or the major compounds (cedrol, widdrol, and thujopsene at 5, 10, 20, 25, 50, and 100 μM) in the presence of 1 μg/mL LPS for 24 h. Curcumin was a positive control at 2.5 and 5 μg/mL concentrations. Following treatment, cell culture supernatants were collected for NO measurement using the Griess reaction. Briefly, 100 μL of cell culture supernatant was mixed with an equal volume of Griess reagent (1% sulfanilamide, 0.1% naphthylethylenediamine dihydrochloride, and 2.5% phosphoric acid) and incubated at room temperature for 15 min. The absorbance was measured at 540 nm using an ELISA microplate reader.NO inhibition (%) = (1 − (sample induced OD_540_ − sample control OD_540_)/(induced OD_540_ − control OD_540_)) × 100

### 3.5. Cytotoxicity Assay

Cytotoxicity was evaluated using the MTT assay on RAW 264.7 macrophages and HepG2 hepatocellular carcinoma cells according to previously described methods for natural compound testing, with minor modifications for Jm-EO [36,37]. For RAW 264.7 cells, 1 × 10^5^ cells/well were seeded in 96-well plates and allowed to attach overnight. The cells were then treated with various concentrations of Jm-EO (1, 5, 10, 20, and 40 μg/mL) for 24 h. Additionally, the major components (cedrol, widdrol, and thujopsene) were tested at concentrations ranging from 5 to 100 μM for 24 h. HepG2 cells (1 × 10^4^ cells/well) were seeded in 96-well plates overnight and then treated with Jm-EO at concentrations of 5, 10, 20, 40, and 80 μg/mL for 24 and 48 h. Plumbagin (1 and 2 μM) was a positive control in the HepG2 experiments. Following the respective treatment periods, 20 μL of MTT solution (5 mg/mL in PBS) was added to each well. The plates were incubated for 1 h (RAW 264.7 cells) or 4 h (HepG2 cells) at 37 °C. The medium was carefully removed, and the formazan crystals were dissolved in 150 μL of DMSO. The absorbance was measured at 570 nm using a microplate reader. Cell viability was calculated as a percentage relative to the untreated control cells.

### 3.6. Antifungal Activity Assay

The antifungal activity of Jm-EO was evaluated against four wood-decay fungi: *Fomitopsis pinicola* (BCRC 3525), *Trametes versicolor* (BCRC 35253), *Lenzites betulina* (BCRC 35296), and *Laetiporus sulphureus* (BCRC 35305) [38]. The antifungal assay was conducted using the poisoned food technique, where Jm-EO was incorporated into sterilized potato dextrose agar (PDA) medium to achieve final concentrations of 25, 50, 100, and 200 ppm. The test plates were inoculated with a 5 mm diameter mycelial disk taken from the edge of 7-day-old cultures of the respective fungi. Control plates contained only PDA medium without essential oil. As a positive control, didecyldimethylammonium chloride (DDAC, Sigma-Aldrich, St. Louis, MO, USA), a commercially available fungicide, was incorporated into PDA medium at a concentration of 100 μg/mL. All plates were incubated at 25 ± 2 °C until the mycelial growth in the control plates reached the edge of the plate. The experiment was performed in triplicate, and the results were averaged. The antifungal index was calculated using the following formula:Antifungal index (%) = (1 − Da/Db) × 100
where Da represents the diameter of mycelial growth in the experimental plate (cm), and Db represents mycelial growth in the control plate (cm).

### 3.7. Statistical Analysis

All experiments were performed in triplicate, and results are expressed as mean ± standard deviation (SD). Statistical analyses were conducted using GraphPad Prism 10.0 (GraphPad Software, San Diego, CA, USA).

## 4. Conclusions

This study investigated the chemical composition and biological activities of Jm-EO. GC-MS analysis revealed a unique chemical profile characterized by a high abundance of specific sesquiterpenoids, including cedrol (22.87%), widdrol (20.82%), and thujopsene (12.26%), which together constituted over 55% of the total essential oil composition, in contrast to other *Juniperus* species that typically feature high levels of monoterpenes. The Jm-EO showed significant anti-inflammatory activity by inhibiting NO production in LPS-stimulated RAW 264.7 macrophages, with an IC_50_ of 12.9 μg/mL. Further investigation of the three major components revealed that widdrol exhibited the most potent anti-inflammatory effect (IC_50_ = 24.7 μM), followed by thujopsene (IC_50_ = 30.3 μM) and cedrol (IC_50_ = 41.1 μM). Notably, this study is the first to report widdrol’s anti-inflammatory activity, identifying it as a key contributor to the Jm-EO’s overall anti-inflammatory properties. Jm-EO showed dose-dependent cytotoxicity against HepG2 hepatocellular carcinoma cells, with an IC_50_ of approximately 41.5 μg/mL after 48 h of treatment. The essential oil also exhibited significant antifungal activity against four wood-decay fungi, particularly effective against *L. sulphureus*, achieving complete growth inhibition at 100 and 200 ppm concentrations.

These findings highlight the potential of Jm-EO as a source of bioactive compounds with practical applications. The diverse biological activities of Jm-EO, including its anti-inflammatory, cytotoxic, and antifungal properties, can be attributed to its major sesquiterpenoid components. Future research should focus on elucidating these compounds’ precise mechanisms of action, particularly widdrol, and exploring their potential development as pharmaceutical agents or natural preservatives. This study contributes valuable information to understanding Jm-EO and opens new avenues for its use in medicinal and wood preservation industry applications. Further research should investigate the specific molecular mechanisms of widdrol’s anti-inflammatory action and explore in vivo models to validate these findings.

## Figures and Tables

**Figure 1 plants-14-02924-f001:**
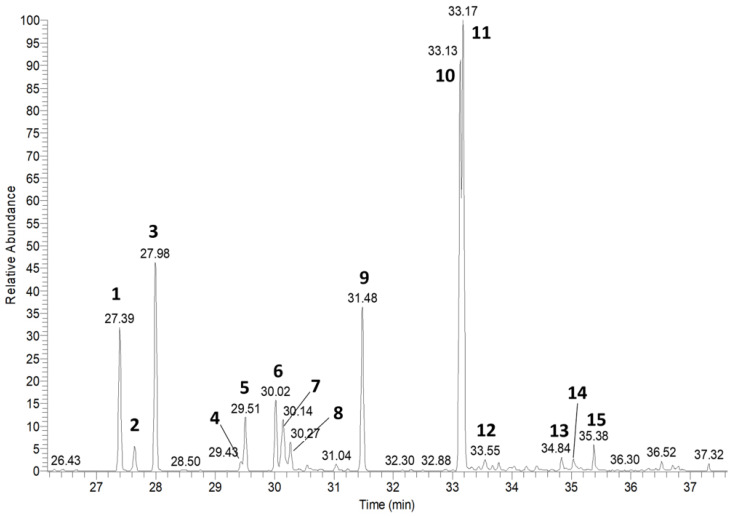
Chromatogram of Jm-EO by GC-MS. 1, *α*-cedrene; 2, *β*-cedrene; 3, thujopsene; 4, *β*-chamigrene; 5, zingiberene; 6, *β*-himachalene; 7, *α*-selinene; 8, cuparene; 9, 8,14-cedranoxide; 10, widdrol; 11, cedrol; 12, 8-cedren-15-ol; 13, 8-cedren-13-ol; 14, valerenol; 15, valerenal.

**Figure 2 plants-14-02924-f002:**
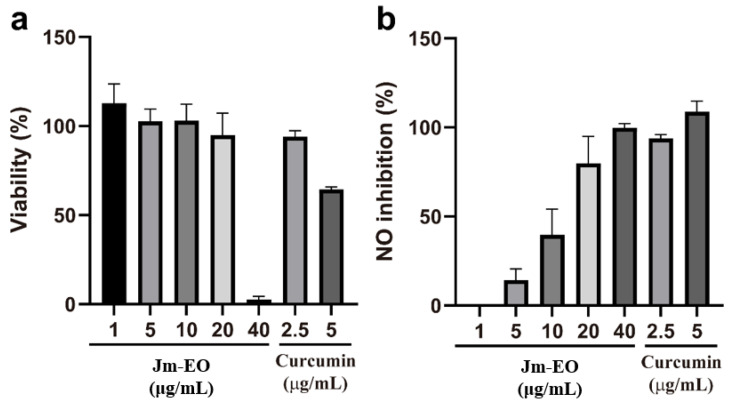
Effects of Jm-EO on RAW 264.7 macrophage cell viability and nitric oxide (NO) production. (**a**) Cytotoxicity of Jm-EO (1–40 μg/mL) and curcumin (2.5 and 5 μg/mL) on RAW 264.7 cells after 24 h of treatment, determined by MTT assay. (**b**) Inhibitory effects of Jm-EO and curcumin on NO production in lipopolysaccharide (LPS)-stimulated RAW 264.7 cells. Cells were treated with LPS (1 μg/mL) in the presence or absence of Jm-EO or curcumin for 24 h. Data are presented as mean ± SD from three independent experiments.

**Figure 3 plants-14-02924-f003:**
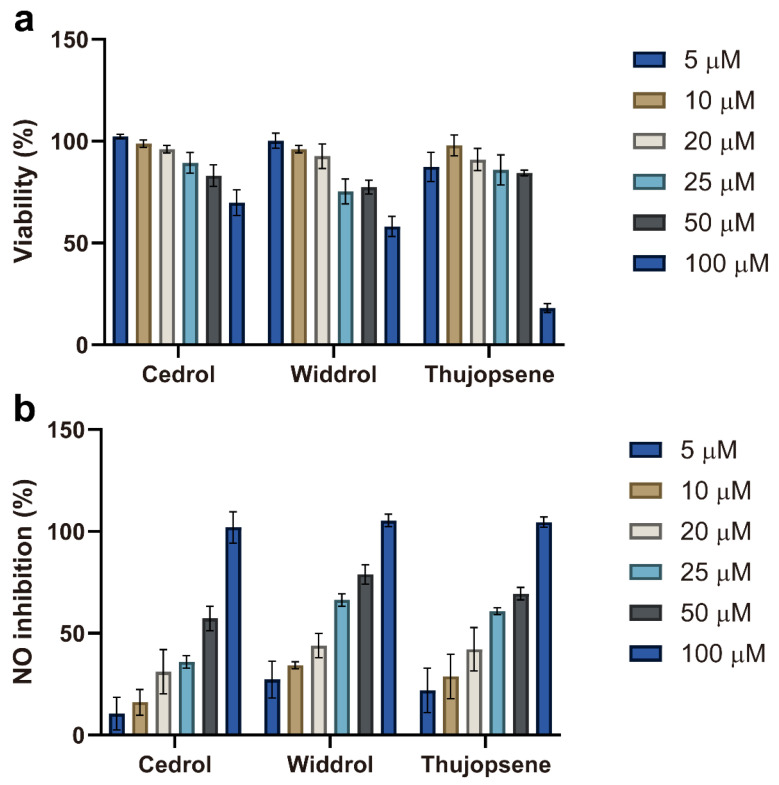
Anti-inflammatory activities of the major components of Jm-EO. (**a**) Cytotoxicity of cedrol, widdrol, and thujopsene (5–100 μM) on RAW 264.7 cells after 24 h, assessed by MTT assay. (**b**) Inhibitory effects of the major components on NO production in LPS-stimulated RAW 264.7 cells. The calculated IC_50_ values were as follows: widdrol (24.7 μM), thujopsene (30.3 μM), and cedrol (41.1 μM). Data are presented as mean ± SD from three independent experiments.

**Figure 4 plants-14-02924-f004:**
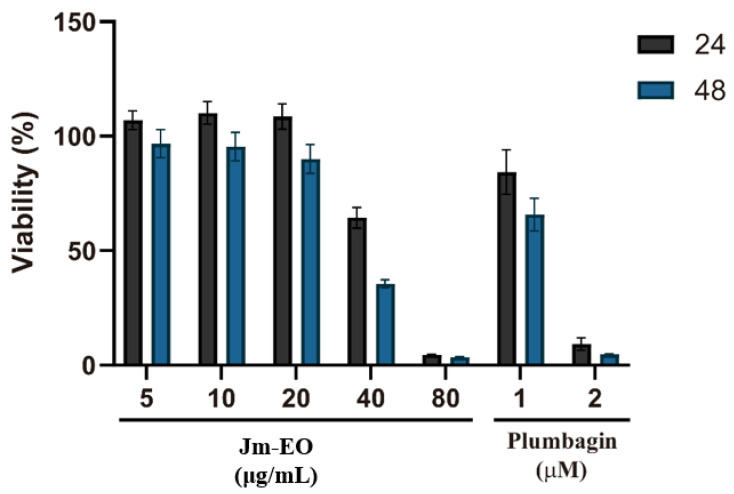
Cytotoxic effects of Jm-EO on HepG2 human hepatocellular carcinoma cells. HepG2 cells were treated with various concentrations of Jm-EO (5–80 μg/mL) or plumbagin (1 and 2 μM) as a positive control for 24 and 48 h. Cell viability was determined using the MTT assay. Jm-EO exhibited dose- and time-dependent cytotoxicity with an IC_50_ value of 52.1 μg/mL (24 h) and 41.5 μg/mL (48 h). Data are presented as mean ± SD from three independent experiments.

**Table 1 plants-14-02924-t001:** Chemical composition of Jm-EO.

RT (min)	Constituent	KI ^a^	Concentration (%)	Reference ^b^
27.39	*α*-cedrene	1415	8.16	MS, KI
27.64	*β*-cedrene	1423	1.51	MS, KI
27.98	thujopsene	1434	12.26	MS, KI, ST
29.43	*β*-chamigrene	1480	0.64	MS, KI
29.51	zingiberene	1482	3.09	MS, KI
30.02	*β*-himachalene	1498	4.12	MS, KI
30.14	*α*-selinene	1502	4.31	MS, KI
30.27	cuparene	1506	1.7	MS, KI
31.48	8,14-cedranoxide	1549	9.5	MS, KI
33.13	widdrol	1605	20.82	MS, KI, ST
33.17	cedrol	1607	22.87	MS, KI, ST
33.55	8-cedren-15-ol	1625	0.87	MS, KI
34.84	8-cedren-13-ol	1686	0.75	MS, KI
35.04	valerenol	1695	0.78	MS, KI
35.38	valerenal	1715	1.18	MS, KI

^a^ Kovats index on the DB-5MS column in reference to *n*-alkanes. ^b^ MS, NIST library; KI, Kovats index; ST, authentic standard compound. In NIST Chemistry Webbook, *α*-cedrene shows in DB-5MS KI of 1418 by Maia, Taveira, et al., 2003; *β*-cedrene in DB-5MS KI of 1423 by Noudogbessi, Yedomonhan, et al., 2008; thujopsene in HP-5MS KI of 1433 by Zhao, Zeng, et al., 2009; *β*-chamigrene in DB-5MS KI of 1481 by Zoghbi, Andrade, et al., 1999; zingiberene in DB-5MS KI of 1484 by Marongiu, Piras, et al., 2006; *β*-himachalene in HP-5MS KI of 1499 by Skaltsa, Demetzos, et al., 2003; *α*-selinene in DB-5MS KI of 1505 by Angioni, Barra, et al., 2006; cuparene in DB-5MS KI of 1504 by Palmeira, Moura, et al., 2004; 8,14-cedranoxide in HP-5MS KI of 1552 by Andriamaharavo, 2014; widdrol in HP-5MS KI of 1597 by Pitarokili, Tzakou, et al., 2006; cedrol in DB-5MS KI of 1607 by Zhu, Li, et al., 2008; 8-cedren-15-ol in DB-5MS KI of 1627 by Cherchi, Deidda, et al., 2001; 8-cedren-13-ol in HP-5MS KI of 1688 by Ghasemi, Asghari, et al., 2003; valerenol in HP-5MS KI of 1711 by Saroglou, Dorizas, et al., 2006; valerenal in HP-5MS KI of 1716 by Zizovic, Stamenic, et al., 2007.

**Table 2 plants-14-02924-t002:** Antifungal activity of Jm-EO.

Jm-EO (ppm)	Antifungal Index (%)			
	*Fomitopsis pinicola*	*Trametes versicolor*	*Lenzites betulina*	*Laettiporus sulphureus*
25	27.68 ± 0.36	13.42 ± 4.81	10.70 ± 0.72	17.73 ± 3.75
50	38.41 ± 1.75	24.01 ± 3.95	12.36 ± 1.90	33.33 ± 6.45
100	48.10 ± 0.43	23.16 ± 5.66	24.40 ± 4.72	100 ± 0
200	63.93 ± 1.13	35.88 ± 1.89	44.34 ± 0.72	100 ± 0
DDAC ^a^ (100 ppm)	100 ± 0	100 ± 0	100 ± 0	100 ± 0

^a^ Didecyldimethylammonium chloride (DDAC) is a commercial fungicide used as a positive control at a concentration of 100 μg/mL.

## Data Availability

The original contributions presented in this study are included in the article. Further inquiries can be directed to the corresponding author.

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
