# Peer review of "Anti-Inflammatory and Antifungal Activities of Wood Essential Oil from Juniperus morrisonicola Hayata"

_plants, 2025, doi:10.3390/plants14182924_

Round 1

Reviewer 1 Report

Comments and Suggestions for Authors

This is an interesting manuscript that explores for the first time the essential oil composition of the tree Juniperus squamata var. morrisonicola, as well as its anti-inflammatory and antifungal properties.

The manuscript is well structured and well written. However, it needs modifications before it can be accepted for publication. Below please see my comments.

  • The scientific name of the species studied must be updated to the current valid name. To do this, it is recommended to consult the website The World Flora Online (https://www.worldfloraonline.org/), created by the highest international authorities on the subject and open access

  • Please spell the word “therapeytic” correctly on line 46

  • I suggest the authors provide a photograph of the tree and the part of the tree from which the extract was prepared.

Author Response

We very appreciate for reviewer’s careful, serious, and constructive comments and suggestions. According to your guidelines, we believe that we have clearly upgraded and improved the quality of our manuscript. Overall, in the revised version, we have almost adapted or addressed the comments and suggestions. The following lists are the special comments that need further response.

Comment 1

The scientific name of the species studied must be updated to the current valid name. To do this, it is recommended to consult the website The World Flora Online (https://www.worldfloraonline.org/), created by the highest international authorities on the subject and open access.

Response 1

We have checked the scientific name of the species according to The World Flora Online (WFO) as suggested. Based on the current taxonomic classification from WFO (ID: wfo-0000355623), the accepted valid name for our study species is Juniperus morrisonicola Hayata, which was first published in Gard. Chron. III, 43: 194 (1908).

Our originally used name "Juniperus squamata var. morrisonicola" is listed as a synonym (WFO ID: wfo-0000735537) in the WFO database. We have updated the manuscript throughout to reflect the current valid nomenclature Juniperus morrisonicola Hayata as recommended by the international taxonomic authorities. The changes have been highlighted in the revised manuscript.

Comment 2

Please spell the word “therapeytic” correctly on line 46

Response 2

We have corrected the word "therapeytic" to "therapeutic". We apologize for this oversight.

Comment 3

I suggest the authors provide a photograph of the tree and the part of the tree from which the extract was prepared.

Response 3

Thank you for this valuable suggestion. While we had already included photographs in our original Graphical Abstract, we have made revisions to better showcase the Juniperus morrisonicola tree and the specific wood parts used for essential oil extraction. The updated Graphical Abstract now provides clearer visual representation of the study species and plant material used in our research.

Reviewer 2 Report

Comments and Suggestions for Authors

The study by Tsao et al, highlight some biological properties of Juniperus squamata var. morrisonicola essential oil.

In the manuscript, the rationale for the research activities and the results obtained are quite clear

Author Response

 Thanks for your careful, serious, and constructive comments and suggestions. According to your guidelines, we believe that we have clearly upgraded and improved the quality of our manuscript. 

Reviewer 3 Report

Comments and Suggestions for Authors

The manuscript of Wang et al. is an interesting experiment showing anti-inflammatory and antifungal activities of essential oil from Juniperus squamata. My suggestions relate mainly to the presentation of results.

The article is well-written, providing detailed results with accuracy. I have a few comments.

Please, in Table 1, I have some suggestions. Change the column “Identification” to “Reference b” and explain the authentic reference from Nist (original article). For example, in Webbook.nist, β-cedrene shows in DB-5 KI of 1419, Adams and Nguyen, 2005, for β-chamigrene Nist showed reference Zoghbi, Andrade, et al., 1999. Make to similar to the others.

Add a chromatogram with peak numbers of the compounds.

Table 2. Please, add 100 μg/mL (100 ppm).

Author Response

Thank you for your helpful and detailed feedback. We have revised the manuscript accordingly to improve transparency and completeness.

Comment 1

Please, in Table 1, I have some suggestions. Change the column “Identification” to “Reference b” and explain the authentic reference from Nist (original article). For example, in Webbook.nist, β-cedrene shows in DB-5 KI of 1419, Adams and Nguyen, 2005, for β-chamigrene Nist showed reference Zoghbi, Andrade, et al., 1999. Make to similar to the others.

Response 1

We have revised Table 1 as suggested. The column "Identification" has been changed to "Reference" and detailed authentic references from NIST Chemistry Webbook have been added to the table footnotes, including the original literature sources, KI values, and analytical columns for each identified compound (lines 80-90).

Comment 2

Add a chromatogram with peak numbers of the compounds.

Response 2

We have added the chromatogram with peak numbers of the compounds as Figure 1 in the revised manuscript.

Comment 3

Table 2. Please, add 100 μg/mL (100 ppm).

Response 3

Thank you for this suggestion. We have added the concentration (100 ppm) to Table 2.